# Determinants of cervical cancer screening among women aged 30 to 49 years in 20 low- and middle-income countries: A multilevel analysis

Mequanent Dessie Bitewa[1]*, Thomas Kidanemariam Yewodiaw[2,3], Aysheshim Asnake Abneh[1], Mikias Getahun Molla[4], Mulat Belay Simegn[1], Tadele Sinishaw Jemere[5], Mequannt Alemu Endayehu[6,7], Aysheshim Belaineh Haimanot[1], Werkneh Melkie Tilahun[1], Atirsaw Assefa Melikamu[4], Tadele Derbew Kassie[1]

**1** Department of Public Health, College of Medicine and Health Sciences, Debre Markos University, Debre Markos, Ethiopia, **2** Department of Epidemiology and Biostatistics, Institute of Public Health, College of Medicine and Health Sciences, University of Gondar, Gondar, Ethiopia, **3** Medical Officer at International Medical Corps, Amhara Region Emergency Operation Center, Gondar Field Office, Gondar, Ethiopia, **4** Department of Public Health, College of Medicine and Health Science, Injibara University, Injibara, Ethiopia, **5** Department of Environmental Health, College of Medicine and Health Science, Wollo University, Dessie, Ethiopia, **6** Department of Epidemiology and Biostatistics, School of Public Health, College of Medicine and Health Science, Bahir Dar University, Bahir Dar, Ethiopia, **7** Public Health Officer at Afro Ethiopia Integrated Development (AEID), Bahir Dar field office, Bahir Dar, Ethiopia

* mequanentdessie30@gmail.com/mequanent_essie@dmu.edu.et

## Abstract

### Background

Cervical cancer is preventable, yet it remains a leading cause of cancer death in women. About 90% of cases and 94% of deaths occur in low- and middle-income countries (LMICs). Limited access to screening drives high incidence and mortality. Screening is central to secondary prevention and global elimination efforts.

### Objective

This study aimed to assess determinants of cervical cancer screening among women aged 30–49 years in low- and middle-income countries: a multilevel analysis.

### Methods

A cross-sectional study used nationally representative data from 148,605 weighted women aged 30–49 years in 20 LMICs (2019–2024). Multilevel logistic regression identified factors associated with cervical cancer screening while accounting for cluster-level variation. Statistical significance was set at $p < 0.05$, with AORs and 95% CIs reported. Model fit and random effects were assessed using deviance, MOR, ICC, and PCV.

**Data availability statement:** The anonymized dataset was available on the Zenodo repository website with a DOI of: https://doi.org/10.5281/zenodo.19437962.

**Funding:** The author(s) received no specific funding for this work.

**Competing interests:** The authors have declared that no competing interests exist.

**Abbreviations:** AOR, Adjusted Odd Ratio; CA,Cancer; CCS, Cervical Cancer Screening; CI, Confidence Interval; DHS, Demographic and Health Survey; HPV, Human Papilloma Virus; ICC, Intra-class Correlation Coefficient; LMICs, Low- and Middle-Income Countries; MOR, Median Odds Ratio; PCV, Proportional Change in Variance; SDG, Sustainable Development Goals; SSA, Sub-Saharan Africa; VIF, Variance Inflation Factor; WHO, World Health Organization.

## Result

Overall cervical cancer screening uptake was 14.03% (95% CI: 13.63–14.45%), ranging from 0.92% in Mauritania to 42.98% in Zambia. Higher screening was associated with older age 40–49 years (AOR = 1.48; 95% CI: 1.41–1.54), occupation (AOR = 1.15; 95% CI: 1.10–1.21), contraceptive use (AOR = 1.38; 95% CI: 1.31–1.44), recent health-facility visit (AOR = 1.93; 95% CI: 1.84–2.02), prior abortion (AOR = 1.28; 95% CI: 1.22–1.34), female-headed households (AOR = 1.11; 95% CI: 1.05–1.18), high community education (AOR = 1.63; 95% CI: 1.49–1.79), and high media exposure (AOR = 2.54; 95% CI: 2.30–2.80). Lower uptake was observed among individuals in high-poverty communities (AOR = 0.63; 95% CI: 0.57–0.68), higher parity (1–4 birth) (AOR = 0.86; 95% CI: 0.78–0.94); (five or more births) (AOR=0.66 95% CI: 59–0.73), and those residing in rural areas (AOR = 0.89; 95% CI: 0.82–0.97).

## Conclusion

Cervical cancer screening uptake in LMICs is far below the WHO 2030 target, with wide country disparities. Socio-demographic factors, health-facility contact, and community education increase uptake, while poverty and geographic barriers reduce it. Integrating screening into routine reproductive and maternal care, strengthening community and media education, and addressing structural barriers to access are essential to improving coverage.

## Introduction

Cancer is characterized by the abnormal growth and spread of cells beyond their usual boundaries, allowing them to invade adjacent tissues and other organs. It can affect any part of the body [1,2]. Cervical cancer, specifically, involves abnormal cell proliferation in the lining of the cervix, typically originating at the squamocolumnar junction [3]. A precancerous cervical lesion refers to an abnormality in cervical cells that may potentially progress to cervical cancer. The progression from a precursor lesion to invasive cancer can take up to 20 years [4].

Globally, cancer is the leading cause of mortality, accounting for one in six deaths in 2022, with nearly 10 million deaths reported [5]. In 2020, the lifetime risk of developing cancer from birth to death was estimated at 25% [6]. Cervical cancer, in particular, was responsible for an estimated 660,000 new cases and 350,000 deaths worldwide in 2022 [7], making it a significant public health concern [8] and one of the leading causes of death among women globally, despite being one of the most preventable cancers [7]. Although it is both preventable and curable, cervical cancer remains the fourth leading cause of cancer-related incidence and mortality in women worldwide [3,7,9]. It is also among the top three cancers affecting women under 45 years of age [9]. Evidence indicates that cervical cancer was the leading cause of cancer incidence in 23 countries and the most common cause of cancer-related

death in 36 countries among women, primarily in Sub-Saharan Africa, South America, and Asia [10]. While incidence and mortality rates are slowly declining, cervical cancer rates remain unacceptably high among women globally [8]. Countries worldwide are striving to eliminate cervical cancer as rapidly as possible in the coming decades, with a set of targets established to be achieved by 2030 [7].

Nevertheless, the incidence and mortality rates of cervical cancer remain significantly above the threshold set by the World Health Organization (WHO), which aims to reduce incidence to no more than 4 cases per 100,000 women-years in all countries, and the disease burden remains high in many regions [8]. Countries with higher development indices have effectively controlled incidence and mortality rates [11], highlighting global inequalities [12]. Mortality rates are 18 times higher in low- and middle-income countries (LMICs) than in developed countries, making the burden particularly higher in LMICs [9]. Some reports indicate that approximately 90% of cases are reported from LMICs [13,14]. According to 2022 data, LMICs accounted for most (94%) cervical cancer deaths, where healthcare access is limited and early detection and treatment are inconsistent [7]. Cervical cancer contributes significantly to the social and economic crises in resource-limited nations; without intervention, deaths due to cervical cancer are projected to increase by 50% [12]. Three out of five cervical cancer cases globally (58%) occurred in Asia, while one out of five cases (20%) were from Africa [8]. Although there has been a decreasing trend in incidence, mortality, and disability worldwide, some countries in Southern Sub-Saharan Africa and Eastern Asia have experienced increases [15].

One of the primary causes of cervical cancer is human papillomavirus (HPV), which is transmitted through sexual intercourse and skin-to-skin contact, infecting the skin and mucous membranes [16]. Getting the HPV vaccine between the ages of 9 and 12 can prevent over 90% of precancerous lesions [17]. Research shows that the highest rate of HPV infection occurs at approximately 25 years of age [18].

Cervical cancer is largely preventable and treatable through HPV vaccination, screening, and early diagnosis, despite its high mortality rate. These are crucial methods for removing CCS in a cost-effective manner [19]. In low- and middle-income countries (LMICs), screening for cervical cancer twice in a lifetime can prevent over 12 million cases, making it an effective strategy to reduce disease incidence. Studies suggest that vaccination, screening, and treatment collectively contribute to significant reductions in the disease burden [19,20]. Twice- or once-lifetime cervical cancer screening would reduce more than one-third (34%) of maternal death by 2030 [20].

The primary purpose of cervical cancer screening is the early detection and treatment of cervical cancer to reduce the disease burden in a cost-effective manner [21]. However, advanced stages require costly procedures and are more challenging to treat [22]. Screening is a crucial step to decrease the risk of dying from cervical cancer [23]. Screening and treatment constitute a secondary prevention approach and represent one of the three key pillars of the global strategy to eliminate cervical cancer [12,24].

To overcome this issue, the WHO has introduced an elimination strategy aiming for 90% early diagnosis and treatment by age 15, 70% screening coverage using a highly effective test at ages 35 and 45, and 90% HPV vaccination coverage by 2030 [8]. The World Health Organization (WHO) advises that adult women aged 30–49 in low- and middle-income countries (LMICs) should have regular cervical cancer screenings [25]. However, some LMICs broaden this recommended age range from 25 to 65 years based on local resources and prevalence of the disease [26,27].

Screening techniques include the Papanicolaou (Pap) smear, HPV DNA testing, and visual inspection with acetic acid (VIA) [3]. VIA is often preferred in low-resource settings because it is affordable, simple, and does not require complex laboratory infrastructure. A positive test result is identified by applying 3–5% acetic acid to the cervix and visually inspecting for aceto-white lesions near the squamocolumnar junction [28].

However, the coverage of cervical cancer screening remains low globally. In 2019, only 36% of women had ever been screened for cervical cancer in their lifetime. Globally, an estimated 67% of women aged 20–70 years, including 64% of women aged 30–39 years, have never been screened for cervical cancer, with lifetime screening coverage ranging from 84% in high-income countries to just 9–11% in lower-middle and low-income countries [29]. Evidence shows that cervical

cancer screening coverage is disproportionately lower in LMICs [30,31]. Lower cervical cancer screening rates are associated with higher incidence and mortality in low- and middle-income countries (LMICs) [32].

Educational attainment, women's autonomy, age, media exposure, and household wealth index are factors associated with cervical cancer screening (CCS) uptake [33]. Some studies have examined region-specific factors, but most have yielded inconsistent results, underscoring the urgent need for large-scale research. Policymakers and program directors must identify the factors influencing cervical cancer screening to develop targeted screening initiatives, reduce morbidity and mortality, and support the WHO's eradication goals. Therefore, this study aims to assess the determinants of cervical cancer screening among women aged 30–49 years in LMICs.

## Methods and materials

### Study area and data source

A cross-sectional study using nationally representative data from 20 LMICs was conducted. The data were obtained after being requested and downloaded from the DHS program website. A total of 27 countries were selected for analysis between 2019 and 2024. Twenty countries (Burkina Faso, Cambodia, Democratic Republic of the Congo, Côte d'Ivoire, Gabon, Ghana, Jordan, Kenya, Lesotho, Madagascar, Mali, Mauritania, Mozambique, Nepal, Nigeria, the Philippines, Senegal, Tajikistan, Tanzania, and Zambia) were eligible for analysis. However, seven countries (Angola, Bangladesh, Gambia, India, Liberia, Sierra Leone, and Rwanda) were excluded due to the lack of CCS data in the dataset. According to the final DHS report of these seven countries, the variable CCS in the questionnaire was not included. The surveys employed multistage cluster sampling designed to collect and present data on key demographic and health indicators. Each survey used a cross-sectional design, collecting data on both independent and outcome variables simultaneously. The DHS program website provided the data for this study in November 2025 upon request through a project submission. The data were accessed via the link https://www.dhsprogram.com/data/dataset_admin.

### Population

The source population consisted of all women aged 30–49 years in low- and middle-income countries (LMICs), and the study population was all women aged 30–49 years in the selected countries.

### Eligibility criteria

All women with an unknown CCS status were excluded.

### Sample size determination and procedure

The DHS employed a two-stage stratified cluster sampling method to select a nationally representative sample, ensuring comparability across regional states within the country. For each survey round, a sampling frame was developed based on the most recent population and housing census. The smallest sampling unit, referred to as an enumeration area or cluster, represents a geographic area containing multiple households and serves as a census counting unit. Clusters were chosen using probability proportional to size sampling in the first stage after rural and urban areas were stratified. In the second stage, households within the selected enumeration areas were chosen through systematic sampling with equal probability. A comprehensive household listing was conducted for each selected cluster. The corresponding DHS country reports contain more details on the sampling technique.

The sample size for our study was all eligible women in 20 LMICs with CCS. Thus, before weighting, 148,403 observations (weighted 148,605) were included in the current study. Country-specific weighted sample sizes ranged from 1,573 in Lesotho to 17,632 in Nigeria (Table 1).

**Table 1. The weighted frequency of women aged 30-49 years in the survey in 20 LMICs DHS surveys from 2019-2024.**

| S. No. | Country | Year of DHS | Frequency | Weighted frequency | Percentage of total sample |
|---|---|---|---|---|---|
| 01 | **Sub-Saharan Africa** | | | **102,263** | **68.82%** |
| 1 | Burkina Faso | 2021 | 7,911 | 7,868 | 5.29% |
| 2 | Cote d'Ivoire | 2021 | 6,855 | 6,642 | 4.47% |
| 3 | Democratic Republic of Congo | 2023−24 | 11,308 | 11,197 | 7.54% |
| 4 | Gabon | 2019−21 | 2,941 | 2,994 | 2.01% |
| 5 | Ghana | 2022 | 7,106 | 7,292 | 4.91% |
| 6 | Kenya | 2022 | 7,635 | 7,592 | 5.11% |
| 7 | Lesotho | 2023−24 | 1,525 | 1,573 | 1.06% |
| 8 | Madagascar | 2021 | 4,097 | 4,094 | 2.76% |
| 9 | Mali | 2023−24 | 7,060 | 7,194 | 4.84% |
| 10 | Mauritania | 2019−21 | 3,300 | 3,272 | 2.20% |
| 11 | Mozambique | 2022−23 | 5,320 | 5,162 | 3.47% |
| 12 | Nigeria | 2023−24 | 17,948 | 17,632 | 11.86% |
| 13 | Senegal | 2023 | 6,856 | 7,023 | 4.73% |
| 14 | Tanzania | 2022 | 6,875 | 6,878 | 4.63% |
| 15 | Zambia | 2024 | 5,849 | 5,850 | 3.94% |
| 02 | **North Africa/ West Asia/Europe** | | | **9,718** | **6.54%** |
| 16 | Jordan | 2023 | 9,482 | 9,718 | 6.54% |
| 03 | **Central Asia** | | | **4,974** | **3.35%** |
| 17 | Tajikistan | 2023 | 5,075 | 4,974 | 3.35% |
| 04 | **South and Southeast Asia** | | | **31,650** | **21.30%** |
| 18 | Cambodia | 2021−22 | 10,567 | 10,842 | 7.30% |
| 19 | Nepal | 2022 | 7,076 | 7,122 | 4.79% |
| 20 | Philippines | 2022 | 13,617 | 13,686 | 9.21% |
| Total | | | 148,403 | 148,605 | 100% |

\* LMICs: low- and middle-income countries

### Study variables

**The Outcome variable** was lifetime CCS uptake, operationalized by the standard DHS questionnaire ("Has a doctor or other health professional ever tested you for cervical cancer?") with a binary response (i.e., yes/no), based on self-reporting of women to have ever undergone CCS.

### Independent variables

**Individual-level factors** include a woman's education (none, primary, secondary, or higher education), occupation (not working or working), age (30–39 years, or 40–49 years) [34], marital status (unmarried or married), media exposure (has or does not have media exposure), contraceptive use (no or yes), permission to seek healthcare (a big problem or not a big problem), number of living children (none, 1–4, or ≥5 children), health insurance coverage (no or yes), health facility visits in the last 12 months (no or yes), history of abortion (no or yes), recent sexual activity (never had sex, active in last 4 weeks, or not active in last 4 weeks), and age at first sex (never had sex, < 15 years, 15–17 years, or ≥18 years).

Factors at the community level encompass place of residence (urban or rural), perceived distance to the health facility (considered a major problem or not), overall community education level (low or high), community poverty status (low or high), community exposure to media (low or high), and the WHO region.

## Data management and analysis

The data were extracted from the most recent DHS datasets 2019 onward, cleaned, recoded, and analyzed using Stata version 17. The data were obtained from standard DHS individual record (IR) datasets, and sample weights (variable v005) were applied to adjust for sampling errors and missing values, thereby restoring the survey's representativeness before conducting statistical analysis. Observations with unknown status and missing CCS values were removed using the drop command. Before appending the datasets, a new cluster identifier variable was generated with unique numbers from each country to avoid overlapping cluster numbers. Frequencies and percentages are presented using tables and graphs.

The assumption of independence was violated in the traditional binary logistic regression because the data used were clustered. Therefore, a multilevel logistic regression model was employed to identify factors associated with CCS and to account for variations within and between clusters. In the final mixed-effects model, p-values less than 0.05 were considered statistically significant, and adjusted odds ratios (AORs) with 95% confidence intervals (CIs) were calculated. These AORs and their 95% CIs served as measures of association in the full model, which examined the relationship between CCS and independent factors at both the individual and community levels.

Four models incorporating variables of interest were fitted to identify factors associated with CCS. The null model, which included no explanatory variables, was used to test for random variation in the intercept and to estimate the intra-class correlation coefficient (ICC). The ICC measures the variation within clusters and is defined as the ratio of between-cluster variance to the overall variance in cervical cancer screening [35].

When comparing two people from two different randomly chosen clusters, the MOR is defined as the median odds ratio between the area with the highest risk and the area with the lowest risk. It was calculated as follows [36]:

$$\text{MOR} = \exp[(\sqrt{2 \times \sigma^2_{between}} \times \phi^{-1} \times (3/4) = \exp\left[\left(\sqrt{2\sigma^2_{between}}\right)(1.112)^{-1} \times \left(1.414 \times \frac{0.75}{1.112}\right)\left(\sqrt{\sigma^2_{between}}\right)\right.$$

$$= \exp[0.9538 \times (\sqrt{\sigma^2_{between}})]$$

Where $\phi^{-1}$ denotes the inverse of the standard normal cumulative distribution function, and $\sigma^2_{between}$ denotes the cluster-level variance in each model.

Model I examined the effects of individual-level characteristics on CCS. Model III (the final model) simultaneously incorporated both individual- and community-level factors to assess their combined effects on CCS. (PCV) is the degree to which quantifies the extent of the unexplained variance of the null model. PCV was calculated using the formula, PCV = (VA-VB/VA) ((VA − VB) / VA) × 100, where VA stands for the variance of the empty model and VB for the variance of the model with extra components.

## Ethical considerations

The ICF Institutional Review Board and the health service ethical review committees of the included countries approved data collection for all DHS surveys. The participants were informed about the advantages and risks of the surveys. Before distributing the Women's Questionnaire, written informed consent was obtained from all eligible participants. DHS got a signed informed agreement from the legally designated representatives (parents or guardians) of teenagers under the age of eighteen. It was a completely voluntary survey. The final datasets did not contain the respondents' names or identification

numbers. Following a formal request, the data of all 27 countries, including an authentication letter, were accessed on November 24, 2025. For this study, no additional authorization was needed. For further details about data and ethical standards, visit https://dhsprogram.com/methodology/Protecting-the-Privacy-of-DHS-Survey-Respondents.cfm.

## Results

### Socio-demographic characteristics

Overall, the study included a weighted sample of 148,605 women aged 30–49 years from LMICs, surveyed between 2019 and 2024. The mean age of the women (mean ± SD) was 38.36 ± 5.64 years, indicating a wide age distribution among the participants. According to Table 2, most women lived in male-headed households (73.60%), and more than three out of five had 1–4 live births (62.31%), while 31.52% had five or more children. Regarding sexual debut, one-fourth of women reported initiating sexual activity before the age of 15 years, and 43.67% initiated it at age 18 years or older. Seven out of ten (70.03%) of women reported no history of contraceptive use. In terms of recent sexual activity, two-thirds (66.03%) were sexually active in the last four weeks. Over half of the participants (52.56%) had visited a healthcare facility in the previous 12 months, and most of the respondents were married (83.37%). More than three-fourths of women had never experienced an abortion (76.66%). Regarding Access-related barriers, 15.75% reported difficulty obtaining permission to seek healthcare, and nearly three-fifths (27.92%) reported distance to a health facility as a significant problem.

At the community level, most women lived in areas characterized by high educational attainment (66.98%), low poverty levels (59.01%), and high media exposure (64.90%). Regionally, the sample was predominantly from Sub-Saharan Africa (68.82%), followed by South and Southeast Asia (21.30%). Just over half of the women (52.06%) resided in rural areas.

Fig 1 presents that the overall CCS uptake in 20 LMICs was 14.03% (95% CI: (13.63–14.45%). Table 2 indicates that across regions, CCS coverage ranges from 12.47% (95% CI: 12.03–12.92%) in SSA to 19.00% (95% CI: 17.29–20.85%) in North Africa/West Asia/Europe. Across countries, it ranges from 0.92% (0.57–1.49) in Mauritania to 42.98% (41.07–44.92%) in Zambia.

Fig 1 shows that some countries had CCS higher than the pooled proportion, such as Lesotho (37.66%), Kenya (27.04%), Gabon (22.10%), Cambodia (22.08%), Burkina Faso (19.48%), Jordan (19.00%), Senegal (15.64%), the Philippines (15.46%), and other countries had CCS of less than the pooled proportion, including Mozambique (12.96%), Tajikistan (12.95%), Tanzania (12.95%), Nepal (11.39%), Côte d'Ivoire (7.75%), Mali (7.32%), Ghana (7.28%), Nigeria (5.00%), Madagascar (2.11%), and DRC (1.07%).

### Associations between explanatory factors and cervical cancer screening

In the final multilevel logistic regression model, which adjusted for both individual and community factors, several variables showed statistically significant associations with cervical cancer screening. Table 3 reveals that women with advanced age had substantially higher odds of being screened compared to younger women. Women aged 40–49 years were 1.48 times as likely to undergo cervical cancer screening (AOR = 1.48; 95% CI: 1.41–1.54).

Women living in female-headed households had 1.11 times higher odds of CCS compared to those in male-headed households (AOR = 1.11; 95% CI: 1.05–1.18). Women using contraceptives were 1.38 times more likely to undergo screening compared to those not using contraceptives (AOR = 1.38; 95% CI: 1.31–1.44). Additionally, employed women had a 15% higher chance of receiving cervical cancer screening than unemployed women (AOR = 1.15; 95% CI: 1.10–1.21). Furthermore, women who perceived obtaining permission to seek healthcare as "not a big problem" were 1.21 times more likely to participate in screening (AOR = 1.21; 95% CI: 1.12–1.30).

Health service–related factors showed strong associations with CCS. Women who had visited a health facility in the previous 12 months were nearly twice as likely to be screened compared to those who had not (AOR = 1.93; 95% CI: 1.84–2.02).

**Table 2. Socio-demographic characteristics of women aged 30-49 years who screened for cervical cancer: evidence from recent surveys in LMICs, DHS from 2019 to 2024.**

| Variables | Category | Cervical cancer screening | | Subtotal (%) |
|---|---|---|---|---|
| | | No | Yes | |
| Respondents' occupation | Not working | 42,970 (87.26) | 6,274 (12.74) | 49,244 (33.14) |
| | Working | 84,779 (85.32) | 14,582 (14.68) | 99,361 (66.86) |
| Women's age | 30 to 39 years | 75,945 (87.21) | 11,143 (12.79) | 87,088 (58.60) |
| | 40-49 years | 51,804 (84.21) | 9,713 (15.79) | 61,517 (41.40) |
| Sex of household head | Male | 94,850 (86.72) | 14,521 (13.28) | 109,371 (73.60) |
| | Female | 32,899 (83.85) | 6,335 (16.15) | 39,234 (26.40) |
| Number of living children | Had no child | 7,961 (86.86) | 1204 (13.14) | 9,165 (6.17) |
| | Had 1–4 children | 77,363 (83.55) | 15,231 (16.45) | 92,594 (62.31) |
| | ≥5 children | 42,426 (90.56) | 4,421 (9.44) | 46,846 (31.52) |
| Age at first sex | never had sex | 10,033 (83.72) | 1,950 (16.28) | 11,984 (8.06) |
| | <15 years | 33,867 (90.27) | 3,649 (9.73) | 37,516 (25.25) |
| | 15-17 years | 29,816 (87.14) | 4,399 (12.86) | 34,215 (23.02) |
| | ≥18 years | 54,032 (83.27) | 10,857 (16.73) | 64,890 (43.67) |
| Contraceptive use | No | 91,733 (88.15) | 12,329 (11.85) | 104,062 (70.03) |
| | Yes | 36,016 (80.86) | 8,527 (19.14) | 44,543 (29.97) |
| Recent sexual activity | Never had sex | 2,204 (95.45) | 105 (4.55) | 2,309 (1.55) |
| | Active in the last 4 weeks | 84,235 (85.75) | 13,887 (14.15) | 98,123 (66.03) |
| | Not active in the last 4 weeks | 41,310 (85.75) | 6,863 (14.25) | 48,173 (32.42) |
| Marital status | Unmarried/separated | 20,858 (84.40) | 3,855 (15.60) | 24,714 (16.63) |
| | Married | 106,891 (86.28) | 17,000 (13.72) | 123,891 (83.37) |
| Visit a health facility in the last 12 months | No | 63,587 (90.20) | 6,609 (9.80) | 70,496 (47.44) |
| | Yes | 64,162 (82.14) | 13,947 (17.86) | 78,109 (52.56) |
| Ever had an abortion | No | 99,050 (86.95) | 14,872 (13.05) | 113,922 (76.66) |
| | Yes | 28,699 (82.75) | 5,984(17.25) | 34,683 (23.34) |
| Getting permission to seek healthcare | Big problem | 21,253 (90.78) | 2,160 (9.22) | 23,413 (15.75) |
| | Not a big problem | 106,496 (85.07) | 18,696 (14.93) | 125,192 (84.25) |
| Distance from health facility | Big problem | 37,102 (89.41) | 4,394 (10.59) | 41,496 (27.92) |
| | Not a big problem | 90,647 (84.63) | 16,462 (15.37) | 107,109 (72.08) |
| Place of residence | Urban | 59,115 (81.74) | 13,202 (18.26) | 72,317 (48.66) |
| | Rural | 68,634 (89.97) | 7,654 (10.03) | 76,288 (51.34) |
| Community educational status | Low | 45,324 (92.38) | 3,740 (7.62) | 49,064 (33.02) |
| | High | 82,425 (82.81) | 17,116 (17.19) | 99,541 (66.98) |
| Community poverty status | Low | 72,098 (82.22) | 15,595 (17.78) | 87,693 (59.01) |
| | High | 55,651 (91.36) | 5,261 (8.64) | 60,912 (40.99) |
| Community media exposure | Low | 47,549 (91.16) | 4,608 (8.84) | 52,157(35.10) |
| | High | 80,201 (83.15) | 16,247 (16.85) | 96,448 (64.90) |
| WHO region | SSA | 86,888 (87.53) | 12,381 (12.47) | 99,269 (**68.82**) |
| | NA/WA/E | 7,871 (81.00) | 1,847 (19.00) | 9,718 (6.54) |
| | Central Asia | 4,330 (87.05) | 644 (12.95) | 4,974 (3.35) |
| | SSEA | 28,661 (82.73) | 5,983 (17.27) | 34,644 (21.30) |

LMICs: low- and middle-income countries; SSA: Sub-Saharan Africa; NA/WA/E: North Africa/West Asia/Europe; SSEA: South and Southeast Asia; WHO: World Health Organization

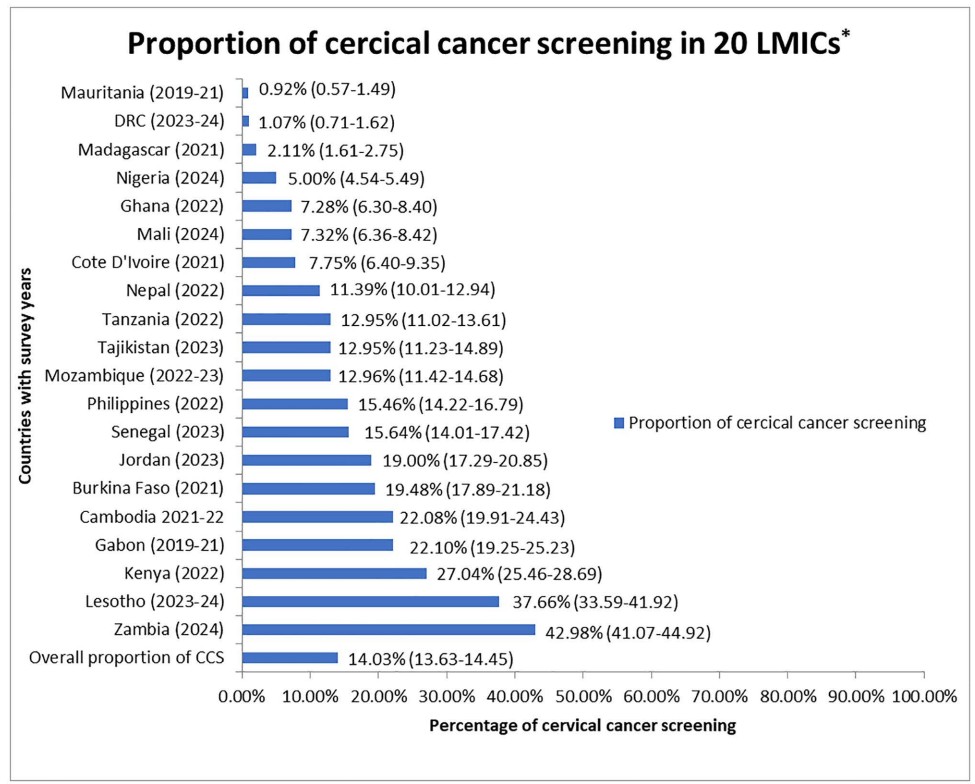

**Fig 1. The proportion of cervical cancer screening among women aged 30 to 49 years in 20 LMICs*: evidence from DHS 2019 to 2024.**
* LMICs- low- and middle-income countries.

Parity and reproductive history also influenced screening behavior. Women with one to four living children were less likely to be screened than those without children (AOR = 0.86; 95% CI: 0.78–0.94), and in women with five or more children, the likelihood of screening reduced by 44% (AOR = 0.66; 95% CI: 0.59–0.73). Women who had ever experienced an abortion had 1.28 times higher odds of screening (AOR = 1.28; 95% CI: 1.22–1.34).

At the community level, contextual factors played a significant role. Women living in communities with high educational attainment had 1.63 times higher odds of undergoing CCS (AOR = 1.63; 95% CI: 1.49–1.79), whereas those residing in high-poverty communities were significantly less likely to be screened (AOR = 0.63; 95% CI: 0.57–0.68). High exposure to community media more than doubled the chances of undergoing screening (AOR = 2.54; 95% CI: 2.30–2.80). Residing in a rural area decreased the likelihood of undergoing cervical cancer screening (CCS) by 10% (Adjusted Odds Ratio = 0.89; 95% CI: 0.82–0.97) compared to living in urban areas. Additionally, individuals from Central Asia and Southeast Asia were 36% (AOR = 0.64; 95% CI: 0.54–0.76) and 11% (AOR = 0.89; 95% CI: 0.81–0.97) less likely, respectively, to participate in screening compared to those from Sub-Saharan Africa (SSA).

## Model fitness and comparison of multilevel models

Finally, the random-effects component indicated substantial clustering of cervical cancer screening (CCS) at the enumeration-area level. Approximately 28% of the total variation in screening uptake was attributable to differences between communities (intra-class correlation coefficient (ICC) = 28.10%). The median odds ratio (MOR = 2.95) indicated considerable heterogeneity in the probability of CCS between clusters, even after accounting for the observed covariates.

**Table 3. Multilevel analysis of factors associated with cervical cancer screening among women aged 30–49 years in 20 LMICs: evidence from DHS 2019 to 2024.**

| Variable | Category | Model I AOR (95% CI) | Model II AOR (95% CI) | Model III AOR (95% CI) |
|---|---|---|---|---|
| **Women's age** | (ref)<br>30–39 years | 1 | – | 1 |
| | 40–49 years | 3.13 (2.97–3.29) | – | **1.48 (1.41–1.54)**\*\*\* |
| **Sex of household head** | Male (ref) | 1 | – | 1 |
| | Female | 1.17 (1.12–1.22) | – | **1.11 (1.05–1.18)**\*\*\* |
| **Contraceptive use** | No (ref) | 1 | – | 1 |
| | Yes | 1.27 (1.22–1.33) | – | **1.38 (1.31–1.44)**\*\*\* |
| **Respondent's occupation** | Not working (ref) | 1 | – | 1 |
| | Working | 1.23 (1.18–1.28) | – | **1.15 (1.10–1.21)**\*\*\* |
| **Marital status** | Not married (ref) | 1 | – | 1 |
| | Married | 1.14 (1.08–1.20) | – | 1.01 (0.95–1.09) |
| **Parity** | Nulliparous (ref) | 1 | – | 1 |
| | 1–4 children | 1.33 (1.25–1.42) | – | **0.86 (0.78–0.94)**\*\* |
| | ≥5 children | 0.88 (0.81–0.95) | – | **0.66 (0.59–0.73)**\*\*\* |
| **Recent health facility visit (last 12 months)** | No (ref) | 1 | – | 1 |
| | Yes | 1.93 (1.86–2.01) | – | **1.93 (1.84–2.02)**\*\*\* |
| **Age at first sex** | Never had sex (ref) | 1 | – | **1** |
| | <15 years | 0.55 (0.48-0.62) | – | 0.63 (0.95-4.16) |
| | 15–17 years | 0.62 (0.55–0.71) | – | 0.72 (0.11–4.76) |
| | ≥18 years | 0.73 (0.65–0.83) | – | 0.81 (0.12–5.34) |
| **Ever had an abortion** | No (ref) | 1 | – | 1 |
| | Yes | 1.32 (1.26–1.37) | – | **1.28 (1.22–1.34)**\*\*\* |
| **Recent sexual activity** | Never had sex (ref) | 1 | – | 1 |
| | Active in the last 4 weeks | 7.59 (6.38–9.01) | – | 5.92 (0.90–38.78) |
| | Not active in the last 4 weeks | 7.54 (6.35–8.95) | – | 5.80 (0.89–7.98) |
| **Getting permission to seek care** | Big problem (ref) | 1 | – | 1 |
| | Not a big problem | 1.21 (1.15–1.28) | – | **1.21 (1.12–1.30)**\*\*\* |
| **Community education** | Low (ref) | 1 | 1 | 1 |
| | High | – | 2.07 (1.93–2.22) | **1.63 (1.49–1.79)**\*\*\* |
| **Community poverty** | Low (ref) | – | 1 | 1 |
| | High | – | 0.66 (0.62–0.71) | **0.63 (0.57–0.68)**\*\*\* |
| **Community media exposure** | Low (ref) | – | 1 | 1 |
| | High | – | 1.75 (1.63–1.88) | **2.54 (2.30–2.80)**\*\*\* |
| **WHO Region** | SSA (ref) | – | 1 | 1 |
| | NA/WA/E | – | 0.56 (0.50–0.64) | 0.86 (0.13-5.73) |
| | Central Asia | – | 0.54 (0.44–0.65) | **0.64 (0.54–0.76)**\*\*\* |
| | South & Southeast Asia | – | 0.67 (0.59–0.77) | **0.89 (0.81–0.97)**\* |
| **Place of residence** | Urban (ref) | – | 1 | 1 |
| | Rural | – | 0.94 (0.88–1.01) | **0.89 (0.82–0.97)**\*\* |
| **Distance to health facility** | Big problem (ref) | – | 1 | 1 |
| | Not a big problem | – | 1.06 (1.02–1.09) | 1.01 (0.95-1.07) |

Table 4 indicates that using deviance (−2 log likelihood), Akaike Information Criterion (AIC), Bayesian Information Criterion (BIC), measures of cluster-level variation (ICC, MOR), and proportional change in variance (PCV), model fitness was evaluated by comparing the null model with three consecutive multilevel logistic regression models. The null model revealed substantial between-cluster heterogeneity, with a cluster-level variance of 2.04 and an intra-class correlation coefficient (ICC) of 38.24%, indicating that more than one-third of the total variation in the CCS was attributable to differences between clusters. The median odds ratio (MOR) of 3.90 further confirmed the presence of significant contextual effects.

Compared to the null model, Model I demonstrated a significant improvement in fit, as evidenced by a substantial reduction in deviance (from 1,714.698 to 792.584) and AIC. The inclusion of covariates decreased the cluster-level variance by 27% (PCV = 27.01%) and reduced the MOR to 3.20, indicating a partial explanation of between-cluster variability. The final model (Model III) exhibited the best overall fit among all models, with the lowest deviance (558.004), AIC (72,528.83), and BIC (72,777.92), reflecting greater precision and explanatory power. The random-effects component revealed substantial clustering of CCS at the cluster level.

In this model, the cluster-level variance decreased to 1.285578, and the intra-class correlation coefficient (ICC) declined to 28.10%, indicating that a substantial portion of the between-cluster variability observed in the null model was explained by the included individual- and cluster-level factors. The median odds ratio (MOR) decreased from 3.90 in the null model to 2.95, reflecting a notable reduction in unexplained contextual heterogeneity. Overall, the final model reduced the between-cluster variance by 36.89% compared to the null model. Multicollinearity was assessed using the mean variance inflation factor (VIF) for Models I to III. The mean VIF values ranged from 1.15 to 1.25 across models, well below the commonly accepted threshold of 10, indicating no evidence of problematic multicollinearity among the explanatory variables.

## Discussion

This study examined the proportion as well as individual- and community-level factors associated with cervical cancer screening (CCS) among women aged 30–49 years across LMICs using multilevel analysis. Overall screening coverage among women aged 30–49 years in LMICs was only 14.03% (95% CI: (13.63–14.45%), which was far below the 70% target set for 2030 [24]. Coverage varies significantly across countries, ranging from 0.92% (95% CI: 0.57–1.49%) in Mauritania to 42.98% (95% CI: 41.07–44.92%) in Zambia. In sub-Saharan Africa (SSA), coverage is approximately 12.47% (95% CI: 12.03–12.92%), and in South and Southeast Asia, it is around 17.27% (95% CI: 16.28–18.31%). Additionally, it was 19.00% (95% CI: 17.29–20.85%) in North Africa/West Asia/Europe. However, only one country (Jordan) was included

**Table 4. Model fit statistics and random-effect measures for multilevel models among women aged 30-49 years in 20 LMICs: based on DHS from 2019-2024.**

| Statistic | Null model | Model I | Model II | Model III (Final) |
|---|---|---|---|---|
| Deviance(−2LL) | 1,714.698 | 792.584 | 1,115.288 | 558.004 |
| AIC | 106,240.9 | 73,707.28 | 104,402.1 | 72,528.83 |
| BIC | 106,260.7 | 73,879.73 | 104,501.2 | 72,777.92 |
| Cluster-level variance | 2.037174 | 1.486916 | 1.651974 | 1.285578 |
| ICC | 0.3824212 | 0.3112798 | 0.334283 | 0.2809733 |
| Median OR(MOR) | 3.90 | 3.20 | 3.41 | 2.95 |
| PCV | Reference | 27.01% | 18.91% | 36.89% |
| Mean VIF | – | 1.15 | 1.19 | 1.25 |

ICC: Intra-class correlation coefficient; VIF: Variance Inflation Factor; AIC: Akaike information criterion; BIC: Bayesian information criterion

in the region. This low coverage may be attributable to cultural and awareness-related factors [37]. Studies indicate that lack of knowledge, poor awareness, economic and cultural barriers, religious restrictions, limited access, lack of trained health workers, and health system challenges are major obstacles to utilizing cervical cancer screening services in LMICs [38,39]. Most LMICs do not have national screening programs and face limited funding, weak infrastructure, and insufficient resources to screen all eligible women [40].

There were significant regional and country-level discrepancies in cervical cancer screening uptake across low- and middle-income countries (LMICs), with the lowest lifetime screening rates observed in sub-Saharan Africa (SSA), Central and South Asian countries [30,41]. Most low-income and lower-middle-income countries still lack official recommendations for cervical cancer screening [29]. Cervical cancer screening programs in many African countries face contextual challenges [42]. In countries such as Mauritania, the Democratic Republic of Congo, Madagascar, Nigeria, Mali, Ghana, Côte d'Ivoire, Nepal, Tanzania, Mozambique, the Philippines, Tajikistan, Senegal, Burkina Faso, Gabon, Cambodia, Jordan, and Kenya, low cervical cancer screening rates reflect structural and health system challenges [37]. The extremely low uptake in Mauritania and the Democratic Republic of Congo is largely attributable to weak health system infrastructure, an inadequate healthcare workforce, poor policy implementation, widespread poverty, and limited availability of screening services [43,44]. In Madagascar, Senegal, Mali, Nigeria, Ghana, Nepal, and the Philippines, low screening coverage is associated with misconceptions, limited transportation, high costs, long travel distances, and substantial infrastructural and geographic barriers [37,45–49]. Additionally, fear of screening and lack of support from husbands or family members can impede participation in screening programs [49]. In Mozambique, Kenya, Tanzania, and Tajikistan, coverage remains constrained not only by infrastructural inequities but also by lack of trained personnel and political barriers, especially in rural areas [50–53].

The proportion of CCS found in this study was lower than the 52.4%, 72%, and 82% reported in studies in France [54], North America [55], and Sweden [56], respectively. This discrepancy may be attributable to variations in socioeconomic factors, especially individual income, as well as differences in the healthcare system's capacity, accessibility, and structure among these regions [57]. The proportion was higher than that reported in a study conducted in Cameroon (4%) [58]. Furthermore, many rural and resource-limited communities remain underserved by healthcare services. Social and economic differences between the populations in the current and previous studies may also explain these discrepancies. The findings of the current study are lower than those reported in studies conducted in various regions worldwide. This discrepancy may be due to the broader geographic scope of our study compared to earlier research [33], which likely encompassed a more diverse range of communities with varying socioeconomic backgrounds.

Furthermore, our analysis used DHS data, whereas other earlier studies used multiple sources; this discrepancy may have led to higher reported screening uptake in those studies. The inclusion of high-income countries in some studies could also explain the relatively higher screening proportions observed [29]. Global estimates derived from pooled data across multiple countries and territories often incorporate high-income settings with near-universal screening coverage, thereby inflating the overall proportion compared to DHS-based estimates, which are limited to low- and middle-income countries (LMICs) [59]. Most European and North American countries have screening coverage exceeding the WHO threshold of 70% [55,60]. These regions benefit from better policies, strategies, and financing for CCS among eligible women [61].

The findings highlight that CCS was associated with socioeconomic, demographic, and health service–related factors. Community education emerged as a strong predictor of screening uptake. In this study, educated women were more likely to undergo CCS. This finding is consistent with other studies [30,33,58,62,63]. Education enhances knowledge and awareness of cervical cancer and its screening, as educated women are more likely to be exposed to information about the disease compared to less educated women. This underscores the importance of implementing educational programs to raise awareness and promote CCS in low-resource settings. Studies indicate that women with higher education levels exhibit better CCS behavior, whereas less-educated women tend to have poor knowledge of cervical cancer and CCS

[64]. Conversely, residing in high-poverty communities is associated with significantly lower screening uptake, reflecting structural barriers such as under-resourced health facilities, indirect costs, and limited outreach services [37]. Women from wealthier households are more likely to undergo cervical cancer screening, a finding supported by previous research [33,58,62,65]. Similarly, women with low socioeconomic status have limited utilization of cervical cancer screening [30], often due to poor knowledge or restricted access to services. People with higher incomes are more likely to seek health care compared to those with lower incomes, and the cost of screening significantly affects their ability to do so [66]. Educational status, place of residence, and wealth collectively contribute to the socioeconomic disparities observed in CCS uptake [67].

Additionally, cultural and awareness-related factors may contribute to lower screening rates. Rural women are less likely to receive CCS compared to urban women, as healthcare access is often limited in remote areas. This was consistent with findings from several studies [30,58,62]. Disparities in socio-demographic, socioeconomic, and behavioral factors further exacerbate this inequality [68].

Exposure to community media was strongly linked to CCS. Women who have greater access to community media are more likely to participate in screening. The finding agrees with findings from other studies [33,69]. Some studies indicate that women who watch television, listen to the radio, or read newspapers at least once a week are more likely to participate in screening tests [70].

Obtaining permission to seek healthcare is a crucial factor influencing cervical cancer screening (CCS). Women who can access care independently or with minimal barriers are more likely to utilize preventive services, including cervical cancer screening [33,49]. This situation likely reflects limited decision-making autonomy and prevailing gender norms that deprioritize preventive reproductive health services. Women from female-headed households are more likely to undergo cervical cancer screening, possibly due to increased autonomy in health-related decisions and greater control over household resources. These women may have greater freedom to seek healthcare services [71]. Additionally, women who make healthcare decisions jointly with their husbands have an increased likelihood of undergoing screening tests [33].

Women older than 40 years are more likely to undergo cervical cancer screening (CCS). As age increases, the likelihood of CCS also increases, which may reflect greater awareness of the benefits of screening among older women [63]. This finding is consistent with studies conducted in Cameroon [58] and sub-Saharan Africa [33,62,63]. Women working in government or private sector jobs were more likely to undergo CCS. This result is consistent with a study from a systematic review and meta-analysis conducted in East Africa [72].

Employed women may have higher income and autonomy, enabling them to access health services more easily. They may be more exposed to information [73]. Health-service contact emerged as a critical facilitator of screening. Women who had visited a health facility in the previous 12 months were more likely to undergo CCS, underscoring the importance of integrating screening into routine primary healthcare services. This is in line with a study in Ghana [68]. The greater opportunities for healthcare providers to offer counseling and health education during routine visits to the health system provide a reasonable explanation for this [74]. To reduce informational and motivational barriers to screening uptake, women who visit health facilities are exposed to qualified health professionals who can dispel misunderstandings, increase awareness about cervical cancer, and directly recommend or offer screening services [75]. The possibility of screening uptake may be lowered by a lack of knowledge about the test [76].

Contraceptive use was significantly associated with higher cervical cancer screening uptake, likely due to more frequent health service contact that enables provider-initiated screening referrals [77]. Studies support our finding [62,78,79]. Parity was another important predictor of the CCS. In contrast to previous findings, women with high parity had a lower likelihood of screening uptake. In a previous study, multiparous and grand multiparous mothers had a higher likelihood of CCS [80]. This could be because women with multiple pregnancies often have regular interactions with healthcare professionals during or after pregnancy. Women with abortion history were more likely to undergo the screening test. They may have more interactions with health providers, which could increase opportunities for screening, counseling, and referrals.

Experiencing an abortion might raise self-awareness of reproductive health risks and motivate uptake of preventive services.

## Limitations of the study

Although DHS data from 20 LMICs provide a large, nationally representative, and standardized dataset that enables strong cross-country comparisons, the data used to determine CCS were based on self-reported screening histories, which are susceptible to recall bias, and did not include clinical results from the screening. Therefore, we were unable to calculate the prevalence of disease among those screened. The cross-sectional nature of the surveys limits causal inference, and these factors may result in underestimation or misclassification. Since we used secondary data, some important variables could not be included due to their absence or missing values, such as the total number of sexual partners, insurance coverage, partners' education, and other behavioral factors.

## Conclusions and recommendations

The CCS uptake in LMICs remains critically low and far below the WHO 2030 target of 70% coverage, with marked cross-country and regional disparities. Every country included in this study has screening rates well under 50%, falling short of the WHO's targets for 2030. Participation in screening is affected by a complex mix of individual characteristics, reproductive background, healthcare accessibility, and broader socioeconomic and informational factors within the community. Age, occupation, contraceptive use, recent health-facility contact, and community education and media exposure were key facilitators, while community poverty and rural place of residence reduced screening uptake.

CCS should be systematically integrated into routine reproductive, maternal, and post-abortion care to leverage existing health-facility contacts, particularly in low-performing countries such as Mauritania, Democratic Republic of Congo, Madagascar, Nigeria, Mali, Ghana, Côte d'Ivoire, Nepal, Tanzania, Mozambique, Philippines, Tajikistan, Senegal, Burkina Faso, Gabon, Cambodia, Jordan, and Kenya, which had less than 30% screening coverage. Community-based education and mass media interventions should be intensified in all WHO regions, where awareness and uptake remain lowest. Structural barriers must be addressed through outreach and mobile screening services, reduced indirect costs, and improved geographic access in underserved and high-poverty communities. Screening programs should prioritize women older than 30 years, while strengthening early awareness among younger women to support timely initiation. Finally, region-specific policy coordination and knowledge transfer from relatively higher-performing countries, such as Lesotho and Zambia, may accelerate equitable progress toward the WHO cervical cancer elimination goals.

## Acknowledgments

The authors would like to thank the DHS program that permitted us to use online data.

## Author contributions

**Conceptualization:** Mequanent Dessie Bitewa, Mequannt Alemu Endayehu, Aysheshim Belaineh Haimanot.

**Data curation:** Mequanent Dessie Bitewa, Aysheshim Asnake Abneh, Mulat Belay Simegn, Tadele Sinishaw Jemere, Mequannt Alemu Endayehu, Aysheshim Belaineh Haimanot, Werkneh Melkie Tilahun, Atirsaw Assefa Melikamu, Tadele Derbew Kassie.

**Formal analysis:** Mequanent Dessie Bitewa, Thomas Kidanemariam Yewodiaw, Aysheshim Asnake Abneh, Mikias Getahun Molla, Mulat Belay Simegn, Mequannt Alemu Endayehu, Werkneh Melkie Tilahun.

**Methodology:** Mequanent Dessie Bitewa, Aysheshim Asnake Abneh, Mikias Getahun Molla, Mulat Belay Simegn, Aysheshim Belaineh Haimanot, Werkneh Melkie Tilahun.

**Project administration:** Mequanent Dessie Bitewa.

**Resources:** Mequanent Dessie Bitewa, Thomas Kidanemariam Yewodiaw, Aysheshim Asnake Abneh, Mikias Getahun Molla, Mulat Belay Simegn, Atirsaw Assefa Melikamu.

**Software:** Mequanent Dessie Bitewa, Thomas Kidanemariam Yewodiaw, Aysheshim Asnake Abneh, Mikias Getahun Molla, Tadele Sinishaw Jemere, Aysheshim Belaineh Haimanot, Werkneh Melkie Tilahun, Atirsaw Assefa Melikamu, Tadele Derbew Kassie.

**Supervision:** Mequanent Dessie Bitewa, Thomas Kidanemariam Yewodiaw, Tadele Derbew Kassie.

**Validation:** Mequanent Dessie Bitewa.

**Visualization:** Mequanent Dessie Bitewa.

**Writing – original draft:** Mequanent Dessie Bitewa, Thomas Kidanemariam Yewodiaw, Aysheshim Asnake Abneh, Mikias Getahun Molla, Mulat Belay Simegn, Mequannt Alemu Endayehu, Aysheshim Belaineh Haimanot, Atirsaw Assefa Melikamu.

**Writing – review & editing:** Mequanent Dessie Bitewa, Thomas Kidanemariam Yewodiaw, Aysheshim Asnake Abneh, Mikias Getahun Molla, Mulat Belay Simegn, Tadele Sinishaw Jemere, Mequannt Alemu Endayehu, Aysheshim Belaineh Haimanot, Werkneh Melkie Tilahun, Atirsaw Assefa Melikamu, Tadele Derbew Kassie.

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
