## [Decision Letter · Decision Letter 0]

16 Mar 2026

Dear Dr. Bitewa

Thank you for submitting your manuscript to PLOS ONE. After careful consideration, we feel that it has merit but does not fully meet PLOS ONE’s publication criteria as it currently stands. Therefore, we invite you to submit a revised version of the manuscript that addresses the points raised during the review process.

**ACADEMIC EDITOR: Please insert comments here and delete this placeholder text when finished.** Be sure to:Be sure to:

After a careful review by the reviewers and the editor, there is a major issues in the methodology of the study. This is the inclusion of the younger age group of less than 30 years to evaluate the determinants of cervical cancer screening and also reporting on the cervical cancer uptake or coverage in this population.  It is noted that the age group according to WHO for cervical cancer screening is over 30 years unless a women is HIV positive. Thus analysis of the study findings should be carefully considered as this may convey a message that is misleading . This especially important as it is noted that there was a higher screening uptake observed in the age group of 30-39 years in the study.It is also important that authors explain how the outcome variable was not included in the countries that were excluded from analysis as this was part of the questionnaire the women answered. How come all women in that country did not answer this question as the criteria to exclude was lack of outcome variable.There is nothing about the 20 LMICs selected for this study in the introduction for the reader to understand why they were selected among others.The introduction does not include the age groups eligible for cervical cancer screening as outlined by WHO and for most LMICs.Add years after each age group or age range throughout the document.The authors should be careful to use the correct verb in the paper e.g severe in page 4 line 91.the DHS country are not available for the reader and therefore to mention that these information is in this document does not help.Please mention the outcome variable instead of just saying the outcome variable in the text.A comprehensive outline of how the study was coordinated throughout the included countries would give clarity to how the study was conducted

Please address all the comments by the reviewers

We look forward to receiving your revised manuscript.

Kind regards,

Ramokone Lisbeth Lebelo, Ph.D

Academic Editor

PLOS One

Journal Requirements:

https://journals.plos.org/plosone/s/file?id=ba62/PLOSOne_formatting_sample_title_authors_affiliations.pdf....

3. In the online submission form, you indicated that [Insert text from online submission form here].

5. We note you have included a table to which you do not refer in the text of your manuscript. Please ensure that you refer to Table 3 & 4 in your text; if accepted, production will need this reference to link the reader to the Table.

Additional Editor Comments (if provided):

Thank you for submitting your paper to PLOS One for possible publication in this journal.

This is an important work on cervical cancer screening, showing the issues and challenges that many countries are facing to eliminate cervical cancer according to WHO 2023 goals.

Reviewers' comments:

Reviewer's Responses to Questions

**Comments to the Author**

1. Is the manuscript technically sound, and do the data support the conclusions?

Reviewer #1: Yes

Reviewer #2: Yes

2. Has the statistical analysis been performed appropriately and rigorously?

Reviewer #1: Yes

Reviewer #2: Yes

3. Have the authors made all data underlying the findings in their manuscript fully available?

Reviewer #1: Yes

Reviewer #2: Yes

4. Is the manuscript presented in an intelligible fashion and written in standard English?

Reviewer #1: No

Reviewer #2: Yes

Reviewer #1: The study area is an important public health problem. It is also informative. The data is presented well but has a lot to work on the English Grammar and typographic errors. The result given on line 339 page19 needs correction.

I wonder how Ethiopia is not included or was not part of the excluded countries in this study? As the authors are from Ethiopia, they should also give emphasis to this point and give further explanation for why it is not part of this study as a SSA nation? It would also be nice if the study can show the prevalence of CxCa in those screened which would give a clue on future decisions? Was there a chance to review a similar study in developed countries for comparison? I didn't see that in your references too. Thank you!

Reviewer #2: Determinants of cervical cancer screening among reproductive-age women in 20 low-

and middle-income countries: a multilevel analysis

The paper is generally well written. The results are clearly spelt out and the discussion well elaborated upon. The study is also topical addressing cervical cancer screening in LMICs that have the highest burden of the disease.

Please find below minor comments:

Line 78- Evidence indicates that cervical cancer was the leading cause of incidence in 23 countries. Should this be “leading cause of cancer incidence”

Line 107- Is a one sentence paragraph. This should not be. Also, that sentence is not properly linked with the following sentences. It appears disjointed.

Entire introduction-The introduction has a lot of information on cervical cancer and cervical cancer screening, which is good. However, the introduction should be presented in such a way that it follows a logical flow and is not just a mix up of ideas.

Line 144- Seven countries were excluded due to the outcome variable. Please state what was the issue with the outcome variable. What is absent in the dataset, or it has a lot of missing data.

Line 151 – The choice of the study population is women aged 15 to 49 years. While this is commonly seen in literature, no cervical cancer screening guideline states screening should start at the age of 15years. WHO recommendation is 30 years while some high-income countries start at 21 or 25 years. Please indicate in the limitations that the uptake/prevalence of screening reported by your study could be lower than the actual due to the inclusion of very young women in the study sample.

Line 170 (Table 1) – Format the table properly. The first row should have the following headings: Country, Year of DHS, Frequency, Weighted frequency, Percentage of total sample.

Table 1 – Nigeria DHS is 2023-24 not 2024.

Line 175 – “Cervical cancer screening with a binary response (i.e., yes/no) was based on self-reporting of women to have ever undergone a cervical cancer screening within five years of the survey” Please cross check how the question was asked. Was it “EVER had cervical cancer screening” or “Had cervical screening in THE PAST 5 YEARS”. Check for all 20 countries.

Line 245 (Table 2) – Format the table better. For the first column, the variable name should fit into only one cell, and not two or three cells. Do same for Table 3.

Limitation- Why do you have only one country in NA/WA/Europe.

Line 427 – Change Young mothers to young women.

Line 443 – Change married mothers to married women. Correct all such changes in the discussion.

Line 448 – “ Contraceptive use is another significant factor for receiving CCS in this study.” Please provide more explanation.

Line 450 – “Agree with other findings (74), multi-parous and grand multiparous mothers had a higher likelihood of CCS.” Please rephrase to improve the grammar.

.

Reviewer #1: **Yes:** Netsanet MengistuNetsanet MengistuNetsanet MengistuNetsanet Mengistu

Reviewer #2: **Yes:** Tope OlubodunTope OlubodunTope OlubodunTope Olubodun

---

## [Author Response · Author response to Decision Letter 1]

6 Apr 2026

A point-by-point response to PLOS ONE

Manuscript ID: PONE-D-25-68229

Title: Determinants of cervical cancer screening among reproductive-age women in 20 low- and middle-income countries: a multilevel analysis

ACADEMIC EDITOR COMMENTS

Comment 1: After a careful review by the reviewers and the editor, there is a major issue in the methodology of the study. This is the inclusion of the younger age group of less than 30 years to evaluate the determinants of cervical cancer screening and also reporting on the cervical cancer uptake or coverage in this population. It is noted that the age group according to WHO for cervical cancer screening is over 30 years unless a woman is HIV positive. Thus, analysis of the study findings should be carefully considered as this may convey a message that is misleading. This is especially important as it is noted that there was a higher screening uptake observed in the age group of 30–39 years in the study.

Response: We sincerely thank the Editor for highlighting this critical methodological concern. We have dropped the age category of less than 30 years, according to WHO guideline,

Comment 2: It is also important that authors explain how the outcome variable was not included in the countries that were excluded from analysis as this was part of the questionnaire the women answered. How come all women in that country did not answer this question as the criteria to exclude was lack of outcome variable.

Response: Thank you for this important clarification request. We have revised the Methods section to clearly state that: although DHS surveys use standardized questionnaires, not all countries include the cervical cancer screening module; countries were excluded only when the variable was entirely absent from the dataset, not due to individual non-response. This reflects dataset-level unavailability, not missing responses. This has been stated in the methods section page 7 of the rtacked version, lines 168 – 169.

Comment 3: There is nothing about the 20 LMICs selected for this study in the introduction for the reader to understand why they were selected among others.

Response: We have revised both the introduction and methods sections to clarify that: the 20 LMICs were selected based on availability of recent [2019 onwards] standard DHS data containing cervical cancer screening variables. Inclusion required comparability of key variables across countries. Before 2019 there was a research conducted on CCS in LMICs. [DOI https://doi.org/10.1200/GO.23.00385]

Comment 4: The introduction does not include the age groups eligible for cervical cancer screening as outlined by WHO and for most LMICs.

Response: We have revised the introduction to include WHO a recommendation indicating that routine cervical cancer screening is recommended for women aged 30 - 49 years in the general population.

Comment 5: Add years after each age group or age range throughout the document.

Response: This has been corrected throughout the manuscript.

Comment 6: The authors should be careful to use the correct verb in the paper (e.g., “severe” in page 4 line 91).

Response: The manuscript has undergone comprehensive language editing, and all grammatical and typographical errors have been corrected.

Comment 7: The DHS country data are not available for the reader and therefore mentioning that these information is in this document does not help.

Response: We will upload the DHS data for 20 LMICs as a Supplementary File with the revised manuscript.

Comment 8: Please mention the outcome variable instead of just saying “the outcome variable” in the text.

Response: We have revised the manuscript to explicitly state the outcome variable as:

“cervical cancer screening (CCS)” throughout the document.

Comment 9: A comprehensive outline of how the study was coordinated throughout the included countries would give clarity to how the study was conducted.

Response: We thank the editor for the suggestion. However, this study is a secondary analysis of publicly available DHS data, not a prospective multi-country study that required active coordination across sites. The data were collected and standardized by The DHS Program. We have clarified this in the Methods section by detailing the specific datasets used, the data access procedure, and the analysis steps taken to ensure reproducibility, in line with PLOS ONE’s guidelines.

Review Comments to the Author

Reviewer 1:

Comment 1: The study area is an important public health problem. It is also informative. The data is presented well but has a lot to work on the English Grammar and typographic errors. The result given on line 339 page19 needs correction.

Response: Thank you for your positive assessment of our study's importance. We have thoroughly revised the manuscript to correct grammatical and typographical errors throughout.

Regarding the specific result on line 339 (page 19), because of re-analysis that specific numeric figure has been changed.

Comment 2: I wonder how Ethiopia is not included or was not part of the excluded countries in this study? As the authors are from Ethiopia, they should also give emphasis to this point and give further explanation for why it is not part of this study as a SSA nation?

Response: This is an excellent and important observation. However, the 2016 Ethiopia Demographic and Health Survey did not include questions regarding cervical cancer screening in its core questionnaires (Household, Woman’s, Man’s, Biomarker, or Health Facility), even if our study was limited to 2019 onwards, which included more recent DHS data. Before 2019, there was a study done on CCS by Abila et al, which covered available data from 2010-2019 [DOI https://doi.org/10.1200/GO.23.00385].

Comment 3: It would also be nice if the study can show the prevalence of CxCa in those screened which would give a clue on future decisions?

Response: We agree that knowing the prevalence of cervical cancer or precancerous lesions among screened women would provide valuable information for program planning. However, the Demographic and Health Survey (DHS) data we used relies on self-reporting of screening history and does not include clinical results from the screening, except in some countries, which have incomplete reports in the dataset. Therefore, we are unable to calculate the prevalence of disease among those screened. We have added this as a limitation in the "Limitations" section to be transparent about the scope of our data.

Comment 4: Was there a chance to review a similar study in developed countries for comparison? I didn't see that in your references too.

Response: Thank you for this suggestion. To provide a more comprehensive context, we have reviewed and incorporated literature on cervical cancer screening uptake in high-income countries. We have added a paragraph to the Discussion section that contrasts our findings with those from developed nations, and we have included new references to support this comparison (new references 54-57, lines 427-431 of tracked version).

Reviewer 2:

Thank you for your encouraging and positive feedback on our work. We appreciate your time and expertise in reviewing it.

Comment 1 (Line 78): Should this be “leading cause of cancer incidence”?

Response: Thank you for the correction. We have revised the sentence on line 78 to read: "Evidence indicates that cervical cancer was the leading cause of cancer incidence in 23 countries … among women." In page 3 lines 71-73.

Comment 2 (Line 107): Is a one-sentence paragraph. This should not be. Also, that sentence is not properly linked with the following sentences. It appears disjointed.

Response: Thank you for important comment. We have removed the one-sentence paragraph and integrated its key point into the preceding paragraph to create a more logical flow of ideas. The information is now connected to the subsequent sentences. Page 6 of tracked version, lines 151-152

Comment 3: The introduction has a lot of information on cervical cancer and cervical cancer screening, which is good. However, the introduction should be presented in such a way that it follows a logical flow and is not just a mix up of ideas.

Response: We have restructured the introduction to improve its logical flow. Page 4-7 of tracked version.

Comment 4 (Line 144): Seven countries were excluded due to the outcome variable. Please state what was the issue with the outcome variable. What is absent in the dataset, or it has a lot of missing data.

Response: We have clarified this point in the "Data Source and Sampling" section. The issue was that the question on cervical cancer screening was absent from the survey questionnaire for those seven countries. The revised text now states: "...seven countries (Angola, Bangladesh, Gambia, India, Liberia, Sierra Leone, and Rwanda) were excluded due to the absence cervical cancer screening in the dataset. According to the final DHS report of these seven countries, the variable CCS the questionnaire was not included." Page 7, lines 168-169

Comment 5 (Line 151): The choice of the study population is women aged 15 to 49 years. While this is commonly seen in literature, no cervical cancer screening guideline states screening should start at the age of 15 years. WHO recommendation is 30 years while some high-income countries start at 21 or 25 years. Please indicate in the limitations that the uptake/prevalence of screening reported by your study could be lower than the actual due to the inclusion of very young women in the study sample.

Response: This is a critical methodological point, and we thank the reviewer for raising it. We completely agree. We have dropped age category 15 to 29 and re-analysed the data using age category 30 to 49 years.

Comment 6 (Line 170, Table 1): Format the table properly. The first row should have the following headings: Country, Year of DHS, Frequency, Weighted frequency, Percentage of total sample.

Response: We have reformatted Table 1 as suggested. The column headings now correctly read: "Country," "Year of DHS," "Frequency," "Weighted Frequency," and "Percentage of total sample."

Comment 7 (Table 1): Nigeria DHS is 2023-24 not 2024.

Response: Thank you for your comment. We verified the dataset from the DHS Program website [https://dhsprogram.com/data/dataset_admin/index.cfm], where it is currently labeled as Nigeria 2024. However, we acknowledge that the survey is commonly referred to as the 2023–24 Nigeria DHS. We have revised the manuscript accordingly. Page 9 of tracked version, table 1, 3rd column.

Comment 8 (Line 175): Please cross check how the question was asked. Was it “EVER had cervical cancer screening” or “Had cervical screening in THE PAST 5 YEARS”. Check for all 20 countries.

Response: We have carefully rechecked the survey questionnaires for all 20 countries included in our study from DHS final reports. The specific question used was consistent across all surveys and asked women if they had ever been screened for cervical cancer. The question from all of the countries asked the women as: has a doctor or other healthcare worker ever tested you for cervical cancer? [Est-ce qu'un médecin ou un autre professionnel de santé vous a déjà fait un test de dépistage du cancer du col de l'utérus? /Has a doctor or another health professional ever performed a cervical cancer screening test on you? Which asked from French versions of some included countries].

Comment 9 (Line 245, Table 2 & 3): For the first column, the variable name should fit into only one cell, and not two or three cells. Do same for Table 3.

Response: We have reformatted Tables 2 and 3 to ensure that all variable names are contained within a single cell, improving the overall readability and professional appearance of the tables.

Comment 10: Why do you have only one country in NA/WA/Europe.

Response: The category "Europe/NA/WA" (Europe, North America, Western Asia) in our regional classification contains only one country because, within our sample of 20 LMICs, only Jordan has standard DHS online data after 2019. We excluded all data before 2019, because there was a study done in LMICs on the CCS by Abila et al, which covered available data from 2010-2019 [DOI https://doi.org/10.1200/GO.23.00385].

Comment 11 (Line 427): Change Young mothers to young women.

Response: Young women have been excluded from the study due to re-analysis.

Comment 12 (Line 443): Change married mothers to married women. Correct all such changes in the discussion.

Response: We have reviewed the entire discussion section and replaced the term "mothers" with "women" in all contexts where it refers to the study population.

Comment 13 (Line 448): “Contraceptive use is another significant factor for receiving CCS in this study.” Please provide more explanation.

Response: it has been corrected accordingly.

Comment 14 (Line 450): “Agree with other findings (74), multi-parous and grand multiparous mothers had a higher likelihood of CCS.” Please rephrase to improve the grammar.

Response: We have rephrased this sentence for better clarity and grammar.

We hope that these revisions meet the approval of the reviewers and the editor.

Sincerely,

Mequanent Dessie Bitewa

---

## [Editor Report · Decision Letter 1]

8 Apr 2026

Determinants of cervical cancer screening among women aged 30 to 49 years in 20 low- and middle-income countries: a multilevel analysis

PONE-D-25-68229R1

Dear Dr. Bitewa

We’re pleased to inform you that your manuscript has been judged scientifically suitable for publication and will be formally accepted for publication once it meets all outstanding technical requirements.

Kind regards,

Ramokone Lisbeth Lebelo, Ph.D

Academic Editor

PLOS One

---

## [Editor Report · Acceptance letter]

PONE-D-25-68229R1

PLOS One

Dear Dr. Bitewa,

I'm pleased to inform you that your manuscript has been deemed suitable for publication in PLOS One. Congratulations! Your manuscript is now being handed over to our production team.

Kind regards,

on behalf of

Dr. Ramokone Lisbeth Lebelo

Academic Editor

PLOS One